# Numerical and Experimental Analysis of Shear Stress Influence on Cellular Viability in Serpentine Vascular Channels

**DOI:** 10.3390/mi13101766

**Published:** 2022-10-18

**Authors:** Khemraj Deshmukh, Saurabh Gupta, Kunal Mitra, Arindam Bit

**Affiliations:** 1Department of Biomedical Engineering, National Institute of Technology, Raipur 492010, India; 2Biomedical Engineering, Florida Tech, Melbourne, FL 32901, USA

**Keywords:** serpentine channels, fluid structure interaction, shear stress, cellular viability, mechanotransduction

## Abstract

3D bioprinting has emerged as a tool for developing in vitro tissue models for studying disease progression and drug development. The objective of the current study was to evaluate the influence of flow driven shear stress on the viability of cultured cells inside the luminal wall of a serpentine network. Fluid–structure interaction was modeled using COMSOL Multiphysics for representing the elasticity of the serpentine wall. Experimental analysis of the serpentine model was performed on the basis of a desirable inlet flow boundary condition for which the most homogeneously distributed wall shear stress had been obtained from numerical study. A blend of Gelatin-methacryloyl (GelMA) and PEGDA200 PhotoInk was used as a bioink for printing the serpentine network, while facilitating cell growth within the pores of the gelatin substrate. Human umbilical vein endothelial cells were seeded into the channels of the network to simulate the blood vessels. A Live-Dead assay was performed over a period of 14 days to observe the cellular viability in the printed vascular channels. It was observed that cell viability increases when the seeded cells were exposed to the evenly distributed shear stresses at an input flow rate of 4.62 mm/min of the culture media, similar to that predicted in the numerical model with the same inlet boundary condition. It leads to recruitment of a large number of focal adhesion point nodes on cellular membrane, emphasizing the influence of such phenomena on promoting cellular morphologies.

## 1. Introduction

Small diameter blood vessels followed by upstream vasoconstriction is often prone to secondary stenosis [1]. Hence, regenerated blood vessels will be an appropriate solution for replacing these small diameter vessels (which were affected with secondary stenosis). Various techniques, such as extrusion, electrospinning, thermal-induced phase separation, braiding, hydrogel tubing, and gas foaming had been used for developing artificial blood vessels [2,3,4,5]. Serpentine vascular geometry having a rigid wall has been used in vascular regeneration [6]. Unlike a straight channel blood vessel, both bifurcated and arch-type blood vessels do not facilitate constant and unidirectional blood flow [7]. Instead, the flow becomes turbulent with larger eddies and torsion. Such phenomena encourages formulation and analysis of the flow driven mechanical force distributions within the serpentine vascular network.

Bioprinting methodologies commonly use hydrogels as a cell-laden material as they are able to come into contact with the cells without damaging their viability [8,9,10,11]. Moreover, hydrogels can readily be mixed with cells, while simultaneously allowing for high cell density and homogenous cell distribution throughout the scaffold [12]. The source of the hydrogel is an important factor because of varying chemical and mechanical properties of synthetic and natural polymers [13]. Naturally derived hydrogels, such as GelMA, consist of inherent signaling molecules that promote cell adhesion. On the other hand, hydrogels derived from other organisms such, as alginate, lack these binding sites for cell adhesion and attachment [14]. Natural biomaterials regulate interactions between the cells and the extracellular matrix and provide an excellent environment for the cells to grow, differentiate, and proliferate [13,15]. Since GelMa is a gelatin based bioink, it can also resolve intricate vascular networks and channels that offer endothelial cells with the essential properties of their native environments. Rheological modifiers also called rheological additives are mixed with gelatin based bioinks to enhance the rheology of the resulting bioink, especially with regard to viscosity and yield stress.

The pulsatile and sinusoidal flows have been reported as the simplified form of physiological blood flow in circulatory system [16,17,18]. These inlet velocity profiles represent simulated biomimetic flow conditions for the serpentine vessels [19,20,21,22]. Further, the flow facilitates the seeded endothelial cells with the physiological stress at the luminal wall of the serpentine structure [23]. Limited study can be found dealing with both physiological and sinusoidal flow to generate hemodynamic stress on the annular surface of the straight and bifurcated substrates [24,25,26]. However, such inlet boundary conditions were not considered in a serpentine network with an elastic wall. Previous studies had considered serpentine models independent of the upstream stenosis (or secondary stenosis). The physiological flow of blood had been found to reproduce adequate stress for the seeded endothelial cells on the luminal wall, while the rigid model of the wall compensates the viability of such cells under a shear stress condition [27,28]. This stress was generated due to the flow of culture media through the bioprinted channels. Due to the positioning of endothelial cells, it is more sensitive and responds quickly to the fluid flow driven stress [29]. It had also been reported that generation of vascular growth factors, conservation of blood vessels and migration of endothelial cells were being improved in the presence of modulating wall shear stress [30,31]. Shear stress derived from flow structures were also found to influence interconnected points between the cytoskeleton and extracellular structures of endothelial cells [32,33,34]. However, it was found that elasticity of the vessel wall promotes migratory phenomena of endothelial cells in the presence of lateral stress in the periodic form of lymphatics. Hence, it is very vital to consider the bioprinted serpentine structure with the elastic wall for enhancing the cell viability of seeded cells in the presence of physiological and sinusoidal flow conditions.

Most of the previous studies deal with the systemic development of a 3D scaffold for formulation of straight channel blood vessels. In addition, a perfusion bioreactor has traditionally been used because of its mechanical advantages rather than a rotatory bioreactor. The present study deals with the use of a 3D printer along with a rotatory bioreactor to fabricate a serpentine vascular structure. The use of a rotatory bioreactor facilitates adequate amounts of shear stress for stimulating the growth of human endothelial cells on the vessel walls. Numerical analysis was conducted to enumerate the relationship between flow driven different mechanical parameters and different inlet and wall boundary conditions. Variation of pressure, stress, and shear rate were found influenced by the inlet and outlet boundary conditions as well as by the viscosity of the acting fluid. There is limited data on cellular viability of vascular channels bioprinted using GelMA bioink mixed with PEGDA photoink. Cellular viability of GelMa hydrogels in a 3D cell culture model demonstrated that the hydrogel scaffolds provide a cell promoting environments for mesenchymal stem cells [35]. Experimental characterization of the bioprinted channels has been performed to validate the optimum flow-driven mechanical parameters, while maintaining higher cellular viability.

## 2. Methodology

A 3D model of the serpentine blood vessel was implemented using COMSOL Multiphysics. Results of the numerical models were further used to optimize the media flow parameters: pressure, shear rate, stress, and axial velocity for different inlet flow boundary conditions. These optimized parameters were further utilized for maintaining the printed serpentine channels in a rotary bioreactor. Finally, a microfluidic based chip system was developed to fabricate the serpentine vascular channels.

### 2.1. Numerical Analysis

The serpentine channel model is shown in Figure 1. Fluid flow (with Newtonian viscosity model for blood equivalent) was evaluated for different wall boundary conditions [slip (S) and no-slip (NS)]. To ensure that the vascular serpentine channel model reflects the realistic simulation of in vivo conditions, appropriate relevant boundary conditions were used for numerical analysis. The wall of the serpentine vascular channel was taken as a rigid body, with a no-slip boundary condition [36]. Fluid–structure interaction (FSI) physics was used for simulating the elastic behavior of the serpentine channel model in another study. For the simulation, the wall of the serpentine vascular structure was considered as a moving mesh. The following properties of blood were used for numerical analysis: density = 1066 kg/m^3^, dynamic viscosity = 3.5 × 10^−3^ Pa.s. Young’s Modulus = 0.4 × 10^6^ Pa, Poisson’s ratio = 0.5. GelMA material properties (density = 1020 kg/m^3^, dynamic viscosity = 4.2 × 10^−3^ Pa.s, Young Modulus 3.18 KPa) were used for the modeling of the serpentine wall. Simulations were performed at a reference temperature of 293.15 K. Navier–Stokes equation was used to represent the momentum of the fluid model. Time-dependent partial differential equation of incompressible Navier–Stokes (Equation (1)) was used for the simulation.
(1)$$\rho \frac{\partial u}{\partial t}-\mu \nabla^{2}\;u+\rho \left(u\times \nabla \right)u+\nabla P=0\;$$
(2)∇×u=0
where *u* is the blood velocity and P is the blood pressure, ρ is the blood density and is the blood viscosity. Fluid was considered to be incompressible and hence a continuity equation of incompressible form was used. Linear elastic material properties of solid mechanics and fluid properties of laminar flow were considered for the study. An isotropic solid model was used for both slip and no-slip conditions. A fully coupled option was selected in COMSOL Multiphysics to establish a possible connection between solid mechanics physics and laminar flow physics. In FSI analysis, all of the meshes created were physics controlled and automatically generated at a normal element size. The statistics of mesh for slip and no slip boundary conditions are listed in Table 1.

A transient model was used to realize the transport phenomena of the fluid in a 2D serpentine channel. A MUMPS based direct linear solver was used to solve the model with a damping factor of unity and a recovery damping factor on 0.75. Velocities corresponding to mean arterial pressure: 80 mmHg, 85 mmHg, and 90 mmHg were considered for the study in Microchannel [37,38,39,40].

Three different flow rate based inlet velocity models of sinusoidal flow (SF) [V_1_ = 4.48 mm/min, V_2_ = 4.62 mm/min and V_3_ = 4.76 mm/min] and physiological flow-PF [V_4_ = 4.48 mm/min, V_5_ = 4.62 mm/min and V_6_ = 4.76 mm/min] were considered in the study. The physiological functions (waveform) were derived from 4D-Laser Doppler data of blood flow across a cross-section of sub-clavicular artery of healthy subject. The study was approved by Institute Ethical Committee (NITRR/IEC/2021/12). Obtained tempero-spatial functions were interpolated thereafter using Curve-Fitting MATLAB toolbox version 2018a, Mathworks, India. Thereafter, the interpolated function with lowest RMSE value was selected for the study. The functions corresponding to inlet velocity models for V_1_ to V_6_ are given in Table 2. One complete cycle of both sinusoidal and physiological flow corresponds to one complete cardiac cycle (represented by T). Different study conditions were formulated based on the above boundary conditions: no-slip (wall) and sinusoidal inlet flow, slip (wall) and sinusoidal inlet flow, no-slip (wall) and physiological inlet flow, slip (wall) and physiological inlet flow (as tabulated in Table 3). A no-slip boundary condition will exclusively provide the influence of inlet flow type in the development of flow physics of media inside the serpentine. On the other hand, slip wall boundary condition will simulate the elasticity of the serpentine wall (similar to the natural blood vessel wall).

Three different locations corresponding to positions P_1_ (4, −2.5), P_2_ (5.8, 2.4) and P_3_ (7.7, −2.5) on the neck, abdominal, and rear region, respectively, were identified on the serpentine channel. These three regions are the downstream region immediately after curvilinear orientation of ascending and descending phases of the serpentine structure, which are more prone to Coriolis forces within the serpentine model. Flow parameters (pressure, shear rate, stress) over the entire cardiac cycle were evaluated at an interval of T/3, T/2, and T cycles.

#### Grid Convergence Test

A grid convergence test was conducted for optimizing the grid size of the flow domain. Grid Convergence Index (GCI) provides a uniform measure of convergence for grid refinement study [41]. The discretization of sinusoidal flow for velocity V_1_ with no slip condition had been selected for three different element sizes. For all element sizes—pressure, shear rate and velocity at probe points P_1_ and P_2_ (as marked in Figure 1) were measured for the entire time cycle. In the current study, the grid refinement ratio *r*, which is equivalent to mean refinement ratio $r_{mean}$of $r_{12}$and $r_{23}$were calculated using Equation (3a)
(3a)$$r=r_{mean}=\frac{r_{12}+r_{23}}{2}$$
where,
$$r_{12}=\sqrt{\frac{\mathrm{element}\;\mathrm{value}\;\mathrm{of}\;\mathrm{Fine}\;\mathrm{discretization}\;}{\mathrm{element}\;\mathrm{value}\;\mathrm{of}\;\mathrm{Normal}\;\mathrm{discretization}\;}\;}$$
$$r_{23}=\sqrt{\frac{\mathrm{element}\;\mathrm{value}\;\mathrm{of}\;\mathrm{Normal}\;\mathrm{discretization}\;}{\mathrm{element}\;\mathrm{value}\;\mathrm{of}\;\mathrm{coarse}\;\mathrm{discretization}\;}\;}$$

Richardson extrapolation [42] introducing the p-th order method, as shown by Equation (3b):(3b)$$p=ln\frac{\left(f_{3}-f_{2}\right)}{\left(f_{2}-f_{1}\right)}\;/\;ln\left(r\right)$$where $f_{1\;,}f_{2}$ and $f_{3}$ are magnitude of derived parameters (e.g., velocity) at point P_1_ and P_2_ at different grid resolution at time T.

With the different parameters listed in Table 4, values of *r* and *p* were evaluated from Equation (3a,b), respectively, and the calculated grid convergence magnitude for two probe points is listed in Table 5. Further, the Grid Convergence Index (GCI) was evaluated using the Richardson extrapolation method based on estimated fractional error [41], as shown in Equation (4a):(4a)$$GCI=\frac{F_{s}\left|e\right|}{\left(r^{p}-1\right)}\times 100\;\%$$

The safety factor (*F*_*s*_) selected for this study was considered to be 1.25.

It was observed that GCI for the fine grid ($GCI_{12}$) is relatively low if compared to the coarse grid ($GCI_{23}$), which demonstrates that the numerical simulation dependency on the cell size was minimum with a decrease in grid size. Figure 2 shows the plot between extrapolated value and the Richardson extrapolation estimation for centerline velocity. The y-axis of Figure 2 represents ${ƹ}_{i+1,i}$, which is the difference in the values of a particular parameter at two different grid resolution and given by:(4b)$${ƹ}_{i+1,i}=f_{i+1}-f_{i}\;$$

It was observed that the extrapolated values for the velocity at point P_1_ and P_2_ did not change significantly on further decreasing the mesh size to represent finer grid resolution.

### 2.2. Experimental Analysis

A co-axial nozzle configuration of a 3D printer was used to print the serpentine channel. The printing bed had been located inside closed chamber housing, as shown in Figure 3. The construct was printed using a blend of GelMA photoInk purchased from CELLINK, Boston USA and PEGDA200 photoInk (purchased from CELLINK, Boston, MA, USA). This photo-ink blend equipped the final construct with cell attachment sites, high shape fidelity, and minimal construct swelling. CaCl_2_ was used as a chemical cross-linker during printing process of the serpentine. The printing environment parameters are presented in Table 6.

#### 2.2.1. Endothelization of Vascular Network

A total of 50–500 µL of HUVEC suspensions (10^7^ cells per mL) was injected via a micro-pipette to fill the channel. Simultaneously, the inlet and the outlet of the construct was sealed with a pinch-clamp. Channels were injected with HUVECs stained with CellTrace CFSE dye (ThermoFisher, Waltham, MA, USA). The device had been centrifuged at 40 rcf for 1 min with a slow acceleration and deceleration to let the cells settle to the bottom. Incubation at 37 °C facilitated the adhesion of cells to form the innermost layer of the vessels. After incubation for 30 min, the construct was placed over 180° for cell adhesion on the other side of the vessel. Finally, the cells were incubated for 5 h at 37 °C on a shaker.

A blood-equivalent working fluid was put in a 10 mL BD Plastipak syringe. The filled syringe was loaded in the programable Shenchen syringe pump and was operated in infusion mode. A syringe pump was connected to microfluidic accessories (one-way luer lock check valve, microfluidic fitting female luer lock adopter kit, barbed to male adopter and syringe tube) in a series to complete the flow driven system. The other end of the tube was connected to a fabricated microfluidic channel placed in the petri dish via a barbed male adopter, microfluidic fitting female luer lock kit, male luer lock kit, fitting adopter male luer to male thread, and mini luer to luer adopter. All the microfluidic components were procured from DARWIN microfluidics.

HUVECS media was fed to the reservoir for cell culture over the printed layers. Regulatory control of each component, such as media pressure, temperature, valve, pH and CO_2_ and N_2_ delivery, etc., were achieved. Induction of media to the housing was made by using a pulsatile pump, as shown in Figure 4. It creates pulsatile hydrostatic pressure to the printed bioink construct, similar to the optimized inlet velocity profiles finalized from numerical study. The printed constructs were imaged using fluorescence microscopy to evaluate cell attachment and cell distribution within the channel.

#### 2.2.2. In-Vitro Testing

Microfluidic channel had been formulated by combining two casted segments. The upper part and the lower part of the printed structure of the model were merged to form a microfluidic system to support a serpentine structure, for housing the printed construct. Media had been transfused through the serpentine structure from the inlet of the structure in the form of sinusoidal waveform and physiological waveform using a programmable syringe pump (DARWIN microfluidic Shenchen Syringe pump, Paris, France), as shown in Figure 4.

#### 2.2.3. Uncertainty Analysis

Uncertainty analysis was performed to validate the model characteristics and to determine the shear stress sensitivity. The uncertainty equation of shear stress on the fabricated channel model was represented by Equation (5). Shear stress was generated when the printed serpentine model came into contact with the flowing HUVECS media.
(5)$$\Delta \tau =\sqrt{{\left(\frac{\Delta \tau }{\Delta P}\sigma_{P}\right)}^{2}+{\left(\frac{\Delta \tau }{\Delta Y}\sigma_{Y}\right)}^{2}+{\left(\frac{\Delta \tau }{\Delta V}\sigma_{V}\right)}^{2}}$$

Here, $\sigma_{P}$,$\;\sigma_{V}\;\mathrm{and}$ $\sigma_{Y}$ are the standard deviations of the pressure, velocity, and the strain rate standard deviation, respectively. From Equation (5), the uncertainty value of stress equivalent was evaluated as ±179.36501 N/m^2^.

#### 2.2.4. Cellular Viability Assessment

The viability of the bioprinted vascular channel was evaluated over a period of 14 days by performing LIVE-DEAD assay. The channels were stained with the LIVE-DEAD cell imaging kit (Fisher Scientific, Waltham, MA, USA). Live cells were stained with Calcein-AM (2 μM), the fluorescence was measured at 494–517 nm, and dead cells were stained with ethidium homodimer-1 (4 μM), which was measured at 528–617 nm. The channels were incubated with LIVE-DEAD stain for 30 min and the samples were washed with PBS. Imaging was then performed using a Nikon C1 Confocal fluorescence microscope. Image J 1.53k open source software developed by Wayne Rasband and contributors National institute of Health, USA was used to evaluate the cell viability.

#### 2.2.5. Sensitivity Analysis

The objective of the sensitivity analysis was to obtain the change of output with respect to change in input parameters. Input parameters may be a material property, variation in geometry and loading condition etc. For a proposed serpentine geometry, sensitivity analysis was performed for a no-slip wall condition of physiological flow corresponding to velocity model V_6_. The sensitivity of the serpentine model was calculated using Equation (6)
(6)$$p=\frac{v^{2}\rho }{2}$$
where p is the calculated pressure, v is the maximum velocity (4.76 mm/min), ρ is the blood density 1066 kg/m^3^. Change in the pressure due to change in density can be expressed by the Equation (7) (modified version of Equation (6))
(7)$$∆p=\frac{v^{2}∆\rho }{2}$$
(8a)$$\mathrm{Sensitivity}=\frac{∆p}{∆\rho }=\frac{v^{2}}{2}$$

The analytical value of sensitivity was calculated by Equation (8)
$$\frac{∆p}{∆\rho }=\frac{4.76\times 4.76}{2}$$


Sensitivity(analytical) = 11.32(8b)


The computed sensitivity was calculated from pressure obtained at blood density (1066 Kg/m^3^) and 6% of blood density (997 kg/m^3^) at a point P_1,_ respectively, as given in Equation (9)
$$\mathrm{Sensitivity}{\;}_{\left(\mathrm{computed}\right)}=\frac{∆p}{∆\rho }$$
$$\mathrm{Sensitivity}{\;}_{\left(\mathrm{computed}\right)}=\frac{12050-11300}{1066-997}$$


Sensitivity(computed) = 10.86(9)


The error of sensitivity was calculated between analytical sensitivity and computed sensitivity by below Equation (10)
$$\mathrm{Error}\;{\;}_{\left(\mathrm{sensitivity}\right)}=\left|\frac{\mathrm{Sensitivity}\left(\mathrm{computed}\right)-\mathrm{Sensitivity}\left(\mathrm{analytical}\right)}{\mathrm{Sensitivity}\left(\mathrm{analytical}\right)}\right|$$
$$\mathrm{Error}\;{\;}_{\left(\mathrm{sensitivity}\right)}=\left|\frac{11.32-10.86}{11.32}\right|$$


Error (sensitivity) = 4.06% (10)


At 6% of change in input parameter (density), it produces a minimum error of sensitivity [43]. From the above analysis it was concluded that an independent material property (density) of the flowing fluid influenced the dependent parameters (i.e., pressure) with a 6% change in density of fluid, as shown in Figure 5.

## 3. Results

Numerical simulation was performed for evaluating flow induced mechanical parameters (stress, pressure, shear rate, and velocity) on the inner lumen of the serpentine model. For condition I (slip wall with sinusoidal flow, refer to Table 3), significant variation of pressure, shear rate, velocity, and stress were observed across the entire model, as shown in Figure 6. For velocity models V_1_ and V_3_, there was no variation in the pressure parameter over time. Maximum pressure generation for velocity models V_1_, V_2_ and V_3_ were recorded as 2–5 × 10^5^ Pa. Similarly, minimum pressure generation for different velocity models V_1_, V_2_, and V_3_ were obtained as 0.2 × 10^5^, −1 × 10^5^, and −1 × 10^5^ Pa, respectively. In the velocity model V_2_, pressure generation on point P_1_ at a time T was maximum, compared to other points, as shown in Figure 6a. It has been observed that the shear rate profile was globally constant for all velocity models, ranging from 0.5 × 10^4^ to 4.5 × 10^4^ 1/s. The shear rate results for all velocity models, around points P_1_ and P_3_ at time T was maximum, as shown in the Figure 6b. For all velocity models, it can be concluded that the flow development was linearly correlated with the progression of time within the T cycle, as shown in Figure 6c. The minimum and maximum axial velocity of 2 mm/min at neck region (P_1_) and 20 mm/min in between the abdominal (P_2_) and rear regions (P_3_), respectively, were observed for condition I. From Figure 6d, it had been inferred that the local maximum stress having a magnitude of 18 × 10^−10^ N/m^2^ had been generated for the case of velocity model V_2_ at the end of the cycle in between the abdominal (P_2_) and rear regions (P_3_).

For the sinusoidal flow with no-slip wall boundary condition (condition II), local minimal variation in pressure, shear rate, and axial velocity were analyzed for velocity models V_1_, V_2_, and V_3_ (Figure 7). A significant variation of pressure distribution was recorded for velocity model V_1_ compared to the two other velocity models. For velocity models V_2_ and V_3_, the local maximum pressure equal to 4 × 10^5^ Pa was observed in the neck region (P_1_) of the channel at end of cycle time T, as shown in Figure 7a. Maximum pressure was generated at the time T/2 sec in the periphery of the rear region (P_2_). It was noticed that the value of axial velocity was similar for V_1_, V_2_, and V_3_ velocity models. The maximum velocity was generated at the end of the T cycle for all inlet velocity models. The minimum and maximum axial velocities were observed as 0 and 20 mm/min, respectively, in the neck region (P_1_) and periphery of the rear region (P_3_), as shown in Figure 7c. Negligible variation had been observed in the shear rate for different velocity models in all regions of the serpentine at T/3, T/2, and T sec. The shear rate profiles for velocity models V_1_, V_2_, and V_3_ were similar and its range of magnitude was 0.5–4.5 × 10^5^ 1/s, as shown in Figure 7b.

The contour plot of pressure for velocity models V_4_, V_5_, and V_6_ at a measured time period T/3, T/2, and T sec was evaluated similarly for physiological flow with no slip wall boundary condition (condition III). For each velocity model, it was observed that local maximum pressure (3.5–4 × 10^5^ Pa) developed at the T/3 and T/2 sec at the neck region of the channel, as shown in Figure 8a. For the same condition, the local maximum (at neck region) and minimum (rear and abdominal regions) shear rate value were predicted as 4.5 × 10^5^ 1/s and 0.5 × 10^5^ 1/s, respectively (Figure 8b). The range of local minimum and maximum axial velocity was calculated between 0 to 18 mm/min, respectively (Figure 8c). The velocity contour was found to be higher for periods T/3 and T/2 sec. Maximum velocity had been developed at the neck region of the serpentine channel.

The contour plot of physiological flow with slip wall boundary condition (condition IV) depicts that for velocity model V_4_, the pressure magnitude at T/3 and T/2 sec was minimum (0.2 × 10^10^ Pa). The maximum pressure (1 × 10^10^ Pa) was developed at approximately the periphery of the neck region and the rear region at the end of the T cycle, as shown in Figure 9a. At T/3 and T/2 secs, local maximum pressure was developed near the periphery of P_1_, as shown in Figure 9a. The contour plot of the shear rate was found similar for all velocity models of the physiological flow condition. The minimum shear rate corresponding to V_4_, V_5_, and V_6_ were obtained as 1 × 10^7^, 0.2 × 10^5^, and 0 × 10^5^, respectively. Similarly, the maximum shear rate for V_4_, V_5_, and V_6_ were 8 × 10^7^, 1.8 × 10^5^, and 2.5 × 10^5^, respectively, as shown in Figure 9b. A maximum axial velocity of 200 mm/min was developed on all measured points at T sec, which was extremely high as compared to other conditions shown in Figure 9c. For velocity models V_5_ and V_6_, a maximum pressure of 2.5 × 10^5^ Pa had been observed on the entire channel at T/3 and T/2 sec at the inlet region. For the same period, the local maximum velocity for model V_6_ were found to be higher than that of V_5_ in the neck as well as the periphery of rear region. For the results obtained in Figure 9d, it was observed that the stress decreases with an increase in velocity. Global values of stress for all velocities fluctuated between 2 and 25 × 10^−10^ N/m^2^ in the rear region of the serpentine channel.

After evaluating the magnitude of various mechanical parameters, the printed serpentine channels were imaged every alternate day using fluorescence microscopy to assess cell attachment and cell distribution within the channel. A sample channel image stained with CFSE is shown in Figure 10a, but the CFSE fluorescent dye may still be producing fluorescence even if the cells are dead. So, for longer time periods samples were imaged without CFSE dye. Sample image of the channel and a cross-section of the channel after 14 days is shown in Figure 10b. Analysis of the LIVE-DEAD assay demonstrated that on average 98.5% of the cells were viable after 14 days in the incubator, as shown in Figure 10c, corresponding to the inlet flow boundary condition of 4.62 mm/min for the luminal fluid (or media). Figure 10d, show a live dead assay fluorescence image for an inlet boundary condition of 4.48 mm/min and 4.76 mm/min, respectively. It is evident from Figure 10d, that cell density in the serpentine channel is less than that corresponding to optimized mechanical parameters, as presented in Figure 10c.

## 4. Discussion

The variation in magnitude of stress, pressure, shear rate, and axial velocity of the fluid flowing through the serpentine vascular channels depends on the material properties, orientation of serpentine geometry, and the flow architecture of working fluid at points P_1_, P_2_, and P_3_. For condition I (as given in Table 3), the proposed architecture follows the Hagen–Poiseuille model for a Newtonian fluid where pressure and velocities are proportional to each other [44]. Variation in the axial velocity influences the shear rate distribution in a closed channel flow. The magnitude of the shear rate describes the flow behavior of working fluid inside the serpentine channel. A minimal variation in shear rate has been found due to a nominal variation in the axial velocity with respect to the radius of the serpentine because of considered rigid wall configuration [45]. Variation in axial velocities also influences the downstream generated pressure in serpentine structures. Localized maximal pressure was generated due to chaotic motion of fluid particles. When the axial velocity of a fluid increases, some of the energy used by the random motion particles to follow the fluid direction develops a lower downstream pressure [46]. The developing downstream pressure further gives rise to localized stress. It was also correlated that the velocity enhancement at the downstream region leads to formation of maximal stress. At higher axial velocity, fluid flow moved slowly near the wall due to diffusion and dispersion of fluid particles [47]. Hence, the inlet velocity V_2_ was found to create higher stress than its counterpart velocity V_3_ = Performing transient analysis, the maximum variation in the axial velocity profile was obtained at the end of the full cycle in condition II (see Table 3) due to positive acceleration of fluid particles [48].

For the no-slip wall boundary condition with physiological flow at the inlet of serpentine (as given in Table 3), a small variation compared to the maximum magnitude of the shear rate was obtained near the wall due to rigidity of the wall of the serpentine channel as well as due to negligible variation of axial velocity magnitude [49]. There was minimum fluid velocity near the wall, which produces a non-significant variation of axial velocity near the wall. According to Hagen–Poiseuille model, maximum pressure magnitude was responsible for producing maximum axial velocity in a serpentine vascular model [50]. For a no-slip condition, when fluid comes into contact with the wall of the serpentine channel, there is no relative movement between them, which decreases the stress magnitude [51].

For the slip boundary wall condition with physiological flow at the inlet of the serpentine (as presented in Table 3), the abrupt changes in axial velocity were contoured by the inlet velocity profile (V_4_), which occurred due to slip conditions at the end of the T cycle. When a fluid flow comes into contact with the curvature section of the serpentine, it creates Coriolis force, which causes a transversal slope in the flowing fluid. Resultant interaction between Coriolis force and transversal slope develops a secondary force on the flow cross-section. This secondary force had disseminated to the curvature section, producing a higher axial velocity magnitude [52]. Maximum axial velocity and presence of curvature of proposed serpentine structure were the effective parameters for developing maximum pressure near the neck and rear region of the serpentine structure at the end of T cycle.

The temporo-spatial deviations in flow-derived parameters were found to affect cell viability and functionality, as given in Table 7. For condition I, the overall maximum and minimum deviation was obtained for pressure and shear rate parameters, respectively. The minimum deviation was obtained due to small changes in the measured point’s value. Serpentine structure and different axial velocity magnitude were responsible for producing maximum pressure deviation [53]. The maximum velocity deviation was obtained due to development of Coriolis force by the serpentine structure, which had caused a radial pressure gradient [54]. Increased pressure gradient radially had further influenced the radial flow of the media, thereby enhancing the deviation in shear stress of adjacent regions [55].

For condition II (refer to Table 3), minimum deviation was obtained for the axial velocity on point P_2_ at the end of T/3 cycle. Similarly, the minimum deviation was reported for the pressure on point P_3_ at the end of the T cycle. As the flow approached near the wall, it was converted into a transitional flow due to elasticity of the wall of the serpentine. Hence, the velocity of flowing fluid increases, producing a maximum deviation of velocities [54].

The laminar flow used in the model and the absence of resistivity to the flow of fluid particles were responsible for producing a slight deviation in the velocity profile [56,57]. For condition III, it was observed that the deviation of all parameters had increased with respect to time. The variation in measured parameters had occurred due to the serpentine structure of the model, responsible for inducing the heterogeneity in the nature of the flow [58]. The maximum and minimum deviation was obtained at point P_3_ in boundary condition IV (refer to Table 3) due to the physiological relevance flow in the presence of the elastic wall. Such a flow condition brought a non-stationary axial velocity profile over the period of time [59].

The numerical analysis aided in the selection of optimized values of different biofluid dynamics features, such as axial velocity, shear rate, pressure, and stress of the proposed serpentine blood vessel model for producing an ideal condition for cell viability. High cellular viability was observed even after 14 days of seeding of HUVECs in the serpentine channel maintained in the bioreactor. Cell attachment was almost uniform across the channel as can be observed from the fluorescent microscopy and confocal images.

## 5. Conclusions

A 3D bioprinting system using an extrusion-based technique was developed to fabricate vascular serpentine channels. The optimized perfusion rate of the media through the printed serpentine channels provided a biomimetic micro-niche for proliferation of seeded HUVECs. Dynamic cultured media at the extracellular surface of seeded cells imposed external forces on the membrane in the form of shear stress and its time derivatives on serpentine walls. Such a distributed pattern of extracellular forces induced faster and synchronous mechanotransduction to the focal points of adhesion of these cells. Hence, a variation in cellular proliferative activities was observed in response to the different perfusion rate of media. Therefore, it can be concluded that flow dynamics features, such as axial velocity, shear rate, stress, and pressure played a vital role in reproducing the physiological behavior and protruding adhesion properties of cells under in vitro conditions on the surface of the serpentine plane. Longitudinal stress generated by the flowing fluid also affects the HUVECS cell’s functionality. It was also observed that the modeling of the serpentine with an elastic wall and consideration of physiological flow conditions at the inlet of the model brought maximal dynamicity in the axial velocity. On the other hand, identified local maxima and minima of flow parameters at designated regions of the serpentine were found to be a useful tool for effective seeding of cells for their proliferation over the surface of the serpentine. The minimum deviation of pressure, shear rate, and velocity were obtained for the sinusoidal flow with a slip wall condition. Result of Live-Dead assay showed the viability of HUVECs after 14 days of printing the channels, demonstrating that the obtained value of fluid dynamics parameters in the serpentine channel was helpful in maintaining the cell proliferation inside the bioreactor.

## Figures and Tables

**Figure 1 micromachines-13-01766-f001:**
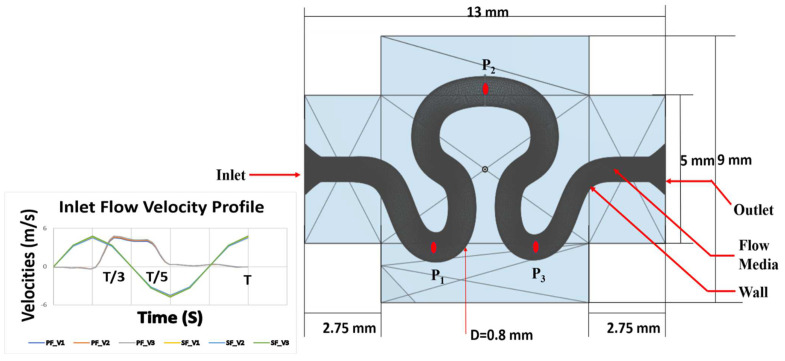
Schematic diagram of the proposed serpentine model.

**Figure 2 micromachines-13-01766-f002:**
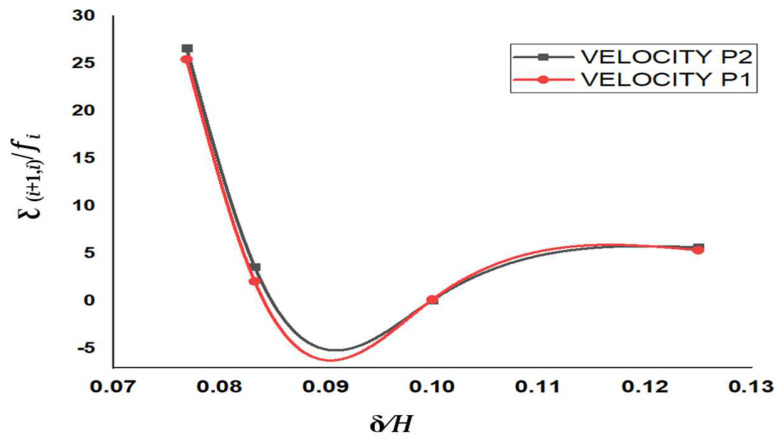
Comparison of velocity variables and extrapolated values between two grid solutions and the Richardson extrapolation estimation.

**Figure 3 micromachines-13-01766-f003:**
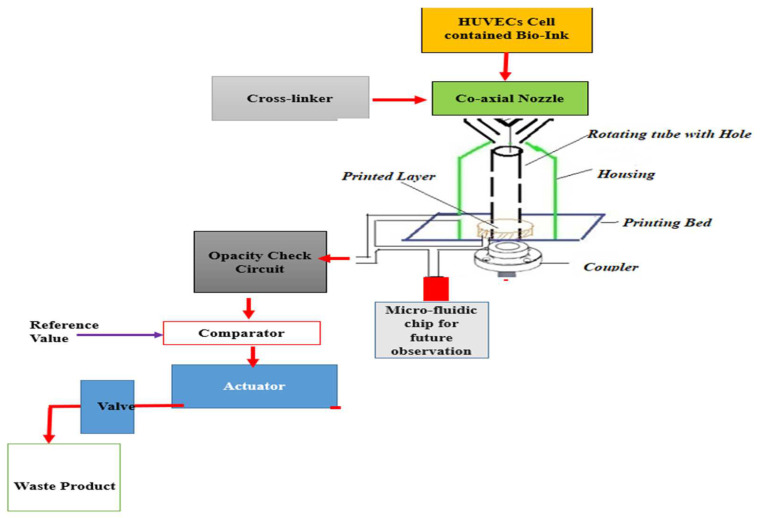
Schematic of 3D printer system for printing of serpentine channels.

**Figure 4 micromachines-13-01766-f004:**
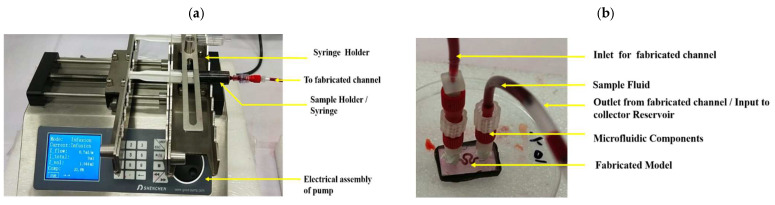
Experimental set up details (**a**) syringe pump setup, (**b**) magnifying image of microfluidic chip.

**Figure 5 micromachines-13-01766-f005:**
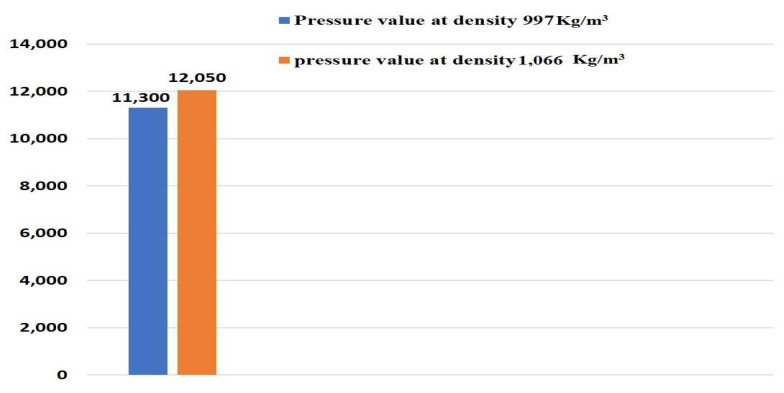
Sensitivity analysis on pressure parameters for two different values of densities.

**Figure 6 micromachines-13-01766-f006:**
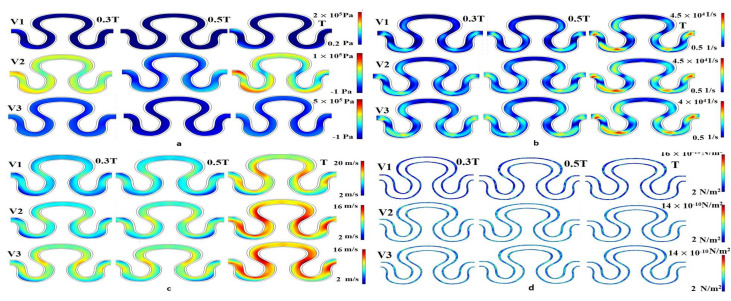
Contour Plot of (**a**) pressure, (**b**) shear rate, (**c**) axial velocity, (**d**) wall stress for sinusoidal flow, slip wall boundary condition for velocity models V1, V2 and V3.

**Figure 7 micromachines-13-01766-f007:**
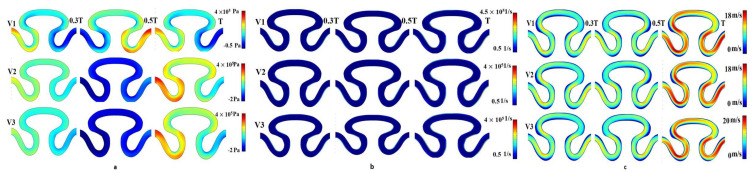
Contour Plot of (**a**) pressure, (**b**) shear rate, (**c**) axial velocity for sinusoidal flow, no-slip wall boundary condition for velocity models V1, V2, and V3.

**Figure 8 micromachines-13-01766-f008:**
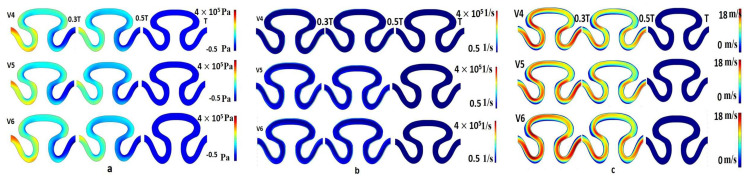
Contour Plot of (**a**) pressure, (**b**) shear rate, (**c**) axial velocity for physiological flow, no-slip wall boundary condition for velocity models V4, V5 and V6.

**Figure 9 micromachines-13-01766-f009:**
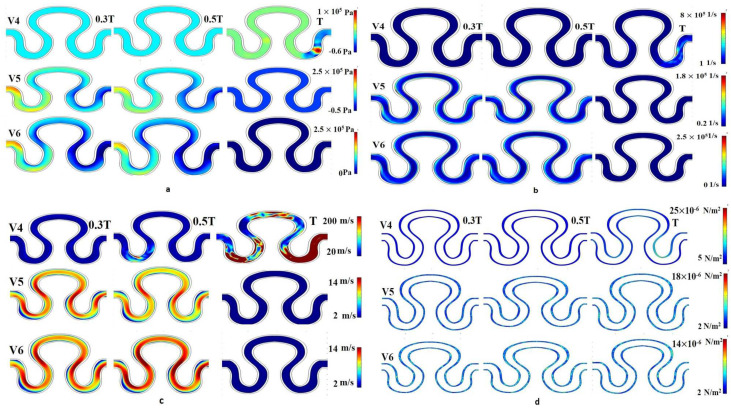
Contour Plot of (**a**) pressure, (**b**) shear rate, (**c**) axial velocity, (**d**) wall stress for physiological flow, slip wall boundary condition for velocity models V4, V5 and V6.

**Figure 10 micromachines-13-01766-f010:**
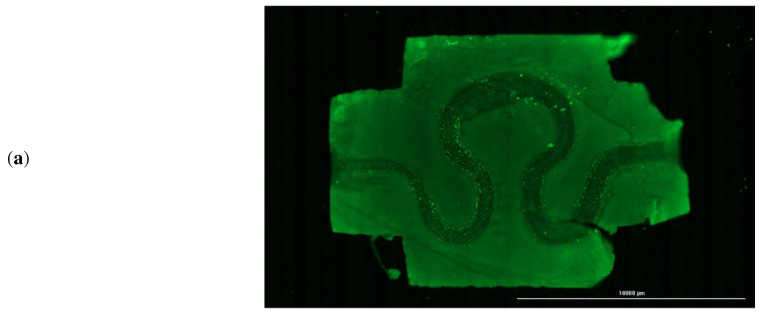
(**a**) Sample bioprinted tissue construct with embedded serpentine channel, (**b**) fluorescent imaging after 14 days, (**c**) confocal imaging after 14 days by staining with LIVE-DEAD assay of serpentine channel seeded with HUVECs at inlet flow velocity = 4.62 mm/min, (**d**) 4.48 mm/min, (**e**) 4.76 mm/min.υ.

**Table 1 micromachines-13-01766-t001:** Mesh statistics.

Parameters	Slip	No Slip
Mesh vertices	2220	1562
Element type	All elements	
Triangles	2608	1578
Quads	712	608
Edge elements	762	328
Vertex elements	40	38
Number of elements	3320	2186
Minimum element quality	0.2758	0.3608
Average element quality	0.7663	0.7733
Element area ratio	0.0761	0.04016
Mesh area	20.31 mm^2^	21.33 mm^2^

**Table 2 micromachines-13-01766-t002:** Velocity function for V_1_ to V_6_.

Inlet Velocity	Model	Parameters Value	Function
Sinusoidal Flow (SF)	V_1_	f = 1.25 Hz, t = 0 to 1 s	V_1_(t) = 4.48× $\sin\left[2\times \pi \times f\times t\right]$
	V_2_		V_2_(t) = 4.62× $\sin\left[2\times \pi \times f\times t\right]$
	V_3_		V_3_(t) = 4.76× $\sin\left[2\times \pi \times f\times t\right]$
Physiological Flow (PF) 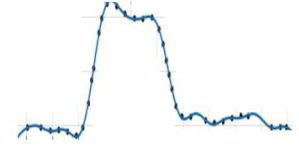	V_4_	t = 0 to 1 s, a_1_ = 3.04, b_1_ = 0.8634, c_1_ = −0.07247, a_2_ = 1.24, b_2_ = 14.02, c_2_ = 2.164, a_3_ = 2.619, b_3_ = 5.81, c_3_ = −0.5393, a_4_ = 0.4497, b_4_ = 24.25, c_4_ = 1.474, a_5_ = 0.3847, b_5_ = 32.5, c_5_ = −2.181, a_6_ = 0.4739, b_6_ = 20.77, c_6_ = 0.4904, a_7_ = 0.2708, b_7_ = 37.65, c_7_ = 2.495, a_8_ = 0.1386, b_8_ = 52.47, c_8_ = −0.7126	V_4_(t) = $a_{1}\times sin\left[\left(b_{1}\times t\right)+c_{1}\right]+\;a_{2}\times sin\left[\left(b_{2}\times t\right)+c_{2}\right]+a_{3}\times \;\;\;sin\left[\left(b_{3}\times t\right)+c_{3}\right]+a_{4}\times \;\;\;sin\left[\left(b_{4}\times t\right)+c_{4}\right]+a_{5}\times \;\;\;sin\left[\left(b_{5}\times t\right)+c_{5}\right]+a_{6}\times \;\;\;sin\left[\left(b_{6}\times t\right)+c_{6}\right]+a_{7}\times \;\;\;sin\left[\left(b_{7}\times t\right)+c_{7}\right]+a_{8}\times \;\;\;\;sin\left[\left(b_{8}\times t\right)+c_{8}\right]$
	V_5_	t = 0 to 1 s, a_1_ = 2.721, b_1_ = 1.588, c_1_ = −0.4734, a_2_ = 1.364, b_2_ = 13.48, c_2_ = 2.332, a_3_ = 3.081, b_3_ = 5.052, c_3_ = −0.1051, a_4_ = 0.3837, b_4_ = 24.37, c_4_ = 1.263, a_5_ = 0.4042, b_5_ = 32.44, c_5_ = −2.133, a_6_ = 0.4619, b_6_ = 19.567, c_6_ = 0.9404, a_7_ = 0.2744, b_7_ = 37.67, c_7_ = 2.482, a_8_ = 0.1435, b_8_ = 52.48, c_8_ = −0.7228	V_5_(t) = $a_{1}\times sin\left[\left(b_{1}\times t\right)+c_{1}\right]+\;a_{2}\times sin\left[\left(b_{2}\times t\right)+c_{2}\right]+a_{3}\times \;\;\;sin\left[\left(b_{3}\times t\right)+c_{3}\right]+a_{4}\times \;\;\;sin\left[\left(b_{4}\times t\right)+c_{4}\right]+a_{5}\times \;\;\;sin\left[\left(b_{5}\times t\right)+c_{5}\right]+a_{6}\times \;\;\;sin\left[\left(b_{6}\times t\right)+c_{6}\right]+a_{7}\times \;\;\;sin\left[\left(b_{7}\times t\right)+c_{7}\right]+a_{8}\times \;\;\;\;\;sin\left[\left(b_{8}\times t\right)+c_{8}\right]$
	V_6_	t = 0 to 1 s, a_1_ = 2.864, b_1_ = 1.405, c_1_ = −0.3165, a_2_ = 1.322, b_2_ = 13.96, c_2_ = 2.122, a_3_ = 3.008, b_3_ = 5.506, c_3_ = −0.3209, a_4_ = 0.7938, b_4_ = 22.63, c_4_ = 2.259, a_5_ = 0.4209, b_5_ = 31.95, c_5_ = −1.982, a_6_ = 0.8909, b_6_ = 20.65, c_6_ = 0.3462, a_7_ = 0.3141, b_7_ = 37.4, c_7_ = 2.577, a_8_ = 0.1474, b_8_ = 52.51, c_8_ = −0.7326	V_6_(t) = $a_{1}\times sin\left[\left(b_{1}\times t\right)+c_{1}\right]+\;a_{2}\times sin\left[\left(b_{2}\times t\right)+c_{2}\right]+a_{3}\times \;\;\;sin\left[\left(b_{3}\times t\right)+c_{3}\right]+a_{4}\times \;\;\;sin\left[\left(b_{4}\times t\right)+c_{4}\right]+a_{5}\times \;\;\;sin\left[\left(b_{5}\times t\right)+c_{5}\right]+a_{6}\times \;\;\;sin\left[\left(b_{6}\times t\right)+c_{6}\right]+a_{7}\times \;\;\;sin\left[\left(b_{7}\times t\right)+c_{7}\right]+a_{8}\times \;\;\;\;\;sin\left[\left(b_{8}\times t\right)+c_{8}\right]$

**Table 3 micromachines-13-01766-t003:** Different boundary conditions for numerical analysis.

Condition	Wall Boundary Condition	Inlet Boundary Condition	Inlet Velocity
I	Slip (S)	Sinusoidal Flow (SF)	V_1_
			V_2_
			V_3_
II	No-Slip (NS)	Sinusoidal Flow (SF)	V_1_
			V_2_
			V_3_
III	No-Slip (NS)	Physiological Flow (PF)	V_4_
			V_5_
			V_6_
IV	Slip (S)	Physiological Flow (PF)	V_4_
			V_5_
			V_6_

**Table 4 micromachines-13-01766-t004:** Generated data for pressure, shear rate, and velocity from simulation.

Probe	Grid	Number of Elements	Pressure at Time T	The Shear Rate at Time T	Velocity at Time T
P_1_	Fine	2366	210,698.8	42,091.81	12.64
	Normal	1578	215,248.60	43,115.82	12.71
	Coarse	1106	230,802.2	44,968.63	12.99
P_2_	Fine	2366	125,274.69	10,222.53	12.66
	Normal	1578	123,677.93	10,938.72	12.25
	Coarse	1106	136,541.84	16,772.07	12.09

**Table 5 micromachines-13-01766-t005:** Grid convergence index for different variables.

Variable	r	*p*	F_s_	$GCI_{23}(\%)$	$GCI_{12}(\%)$
Pressure (P_1_)	1.209	6.48	1.25	3.72	1.11
Shear Rate (P_1_)		3.12		6.6	3.7
Velocity (P_1_)		6.41		1.13	0.3
Pressure (P_2_)		10		22.9	0.28
Shear Rate (P_2_)		9.39		13.48	1.77
Velocity (P_2_)		27		0.003	0.24

**Table 6 micromachines-13-01766-t006:** Parameters used for 3D printing of serpentine channel.

Parameters	Specification
Printing house temperature	37 °C
Extrusion pressure of the coaxial nozzle	0.1 to 1 Mpa
Nozzle diameter	1 mm
Needle size	20 gauge
Extrusion rate	5 mm/s
Dispensing speed	4 mm/s
Printing bed temperature and material	2 °C to 40 °C and Aluminum/Glass
Pulsatile pump speed, pressure and flow rate	5 to 36 rpm, 80 to 120 mmHg; and 1–4 mL/min
pH of media	7.2 to 7.4

**Table 7 micromachines-13-01766-t007:** Calculated deviation for pressure, shear rate and velocity.

Configuration	Parameter	Point	Time	Deviation (%)	
S_SF_V1, V2, V3	Pressure	P3 (V2) percent [V2, V3]	0.5	Min	0.24937
		P2 (V2) percent [V2, V3]	0.3	Max	231.1239
	Shear rate	P2 (V2) percent [V2, V3]	0.5	Min	0.19267
		P2 (V1) percent [V2, V1]	0.3	Max	19.17148
	Velocity	P2 (V3) percent [V2, V3]	0.5	Min	0.324655
		P1(V3) percent [V1, V2]	0.5	Max	36.753
NS_SF_ V1, V2, V3	Pressure	P1 (V3) percent [V2, V3]	0.3	Min	3.603224
		P3 (V1) percent [V3, V1]	1	Max	14.14604
	Shear rate	P1 (V3) percent [V2, V3]	0.3	Min	1.854657
		P2 (V1) percent [V3, V1]	0.5	Max	9.254331
	Velocity	P2 (V3) percent [V2, V3]	0.3	Min	1.775892
		P1 (V1) percent [V3, V1]	1	Max	6.57622
NS_PF_V4, V5, V6	Pressure	P1 (V6) percent [V6, V5]	1	Min	3.075083
		P3 (V4) percent [V6, V4]	0.5	Max	14.18746
	Shear rate	P2 (V5) percent [V4, V5]	1	Min	2.104332
		P2 (V4) percent [V6, V4]	0.5	Max	8.072614
	Velocity	P3 (V6) percent [V6, V4]	1	Min	2.813501
		P1 (V4) percent [V6, V4]	0.5	Max	6.5163
S_PF_ V4, V5, V6	Pressure	P3 (V6) percent [V6, V5]	0.3	Min	2.485286
		P3 (V6) percent [V6, V4]	1	Max	−3.9 × 10^8^
	Shear rate	P3 (V6) percent [V6, V4]	1	Min	2.959357
		P1 (V6) percent [V6, V5]	0.3	Max	3229923
	Velocity	P3 (V4) percent [V4, V5]	0.5	Min	0.7726
		P3 (V5) percent [V4, V5]	1	Max	291,407

## Data Availability

Data will be made available based on request made.

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
