# Peer review of "Numerical and Experimental Analysis of Shear Stress Influence on Cellular Viability in Serpentine Vascular Channels"

_micromachines, 2022, doi:10.3390/mi13101766_

Round 1

Reviewer 1 Report (Previous Reviewer 2)

The authors made an effort to include some of the missing details for both the simulation and the experimental approaches; however, there are still some issues that need to be addressed prior to publication:

1. The sensitivity analysis should be presented graphically and the conclusions in that regard included in that section.

2. The mesh convergence analysis should be presented for various points along the computational domain graphically. This will convince the audience that effectively the simulations are valuable.

3. The conclusions still fail to reflect the novelty of the proposed approach and should be improved.

Author Response

Reviewer 2 Report (New Reviewer)

The authors presented a well written manuscript with the interesting point of demonstrating viability of cells during bioprinting process, using different conditions. The bioprinting method and the mathematics are described quite in depth and certainly very clearly for scientists in the field.  Although the focus and the importance of the work, in my opinion, is about demonstrating cell viability, which unfortunately, has been conducted in a poor manner. The biological readout, in fact, is very confusing and not sufficient. Therefore I consider the manuscript not publishable like it is and I explain here below my concern and suggestion to improve it.

1) The LIVE/DEAD assay is not compared with proper controls ( cells that are supposed to be all alive). The assay must be repeated multiple times to have a significant result and perform statistical analysis.

2) Cells should be stained for the nuclei as well (DAPI) to make sure that the green fluorescence due to CALCEINE AM (Live cells) is really inside of the cells and not a background signal. The use of confocal microscopy has the exact scope to allow colocalization of different fluorescence emissions to detect if the are generate from the same source/location/cell compartment.

3) The use of CFSE is confusing. CFSE and Calceine have the same emission spectra, therefore the green fluorescent is not clear if is because of CFSE or LIVE staining. Are cells staining with LIVE/DEAD kit also pre-stained with CFSE? The general use of CFSE is to trace cell proliferation, therefore the more cells proliferated, the weaker the CFSE signal become...is this the case? I would rather use GFP expressing HUVEC to evaluate the How many cells are in the channels at the different conditions

4) Along with the viability test. Something that would add much more significance to this work and manuscript would be the performance of a functional test of the cells in the channels, like evaluate gene expression or protein expression. Cells can be still alive in the channels but not functional anymore, or they could take some time to recover.

I hope these comments help the author to improve their work and resubmit it

Author Response

Reviewer 3 Report (New Reviewer)

The work presents an approach to simulating, manufacturing, and testing serpentine-based devices that provide a closer representation of in vivo vascular vessels.

1. According to the syringe pump shown in Figure 3, it is not possible to meet the long and stable circulating culture environment of cells in the serpentine flow channel, so how is the cell culture carried out?

2. The article mentions the printing of the sample tissue, but lacks specific printing details. Although the relevant work done in the early stages is mentioned, it is not enough to illustrate the manufacturing process of the printed samples in this article.

Author Response

This manuscript is a resubmission of an earlier submission. The following is a list of the peer review reports and author responses from that submission.

Round 1

Reviewer 1 Report

This work presents COMSOL modelling of a GelMA-PEGDA blend hydrogel channel seeded with HUVECs and subjected to sinusoidal and pulsatile flow patterns designed from clinical data. The authors claim novelty in their approach to modelling hydrogel wall elasticity in their numerical model, identifying optimum flow-driven mechanical parameters resulting in high cellular viability. The latter claim is unsubstantiated since it is unclear how the numerical modelling influenced or identified experimental parameters that improved cell viability. The methodology for assessing cell viability is scientifically concerning, there are several critical missing experimental details which would make these results unreproducible.

The authors have published on the topic of microfluidic models with shear stress modelling (https://doi.org/10.3390/mi13020305), however this work is distinct in its use of a different cell type, microfluidic design and the fluid flow patterning. A computational and experimental simulation of physiological flow rate, derived from patient data has potential to be an informative study, however the objectives of this study and the hypotheses it seeks to test are ambiguous. Moreover, the methodology is not compelling, and the conclusions are not supported by the data presented. The presentation of the data is difficult for readers to follow, and significant editing of the manuscript is required for grammatical errors.

Below are some specific major and minor comments.

1.      Unclear novel contribution. The only sentences in the introduction which indicate what questions this work seeks to answer are:

“Numerical analysis was conducted to enumerate the relationship between flow driven different mechanical parameters and different inlet and wall boundary conditions. Variation of pressure, stress, and shear rate were found influenced by the inlet and outlet boundary conditions as well as by the viscosity of the acting fluid… Experimental characterization of the bioprinted channels has been performed to validate the optimum flow-driven mechanical parameters while maintaining higher cellular viability.”

The second sentence describes well-established fluid dynamics concepts as an original finding. The Authors don’t state what the ‘optimum flow-driven mechanical parameters’ are, and there is no experimental data to validate their claim that the flow parameters they used lead to higher cell viability than the other parameters that were modelled computationally. If the intention had been that the flow rate patterns were the focus of this study, then a uniform flow rate control would be necessary to demonstrate an improvement in cell behavior.

2.      Imbalance of numerical and experimental data. Out of 8 figures, 3 show the experimental configuration, 4 are COMSOL plots and 1 shows a qualitative image of one in vitro experiment. In the results section (lines 180-249), only lines 243-249 provide some experimental details. This is at odds with this statement in the abstract:

The objective of the current study was to perform a numerical and experimental analysis of the influences of different mechanical stimuli: pressure, stress, and shear rate on cellular viability in a serpentine vascular network channel.”

This relevant 2020 paper (DOI: 10.1126/sciadv.abb3629 -which the authors did not cite) similarly investigates the effects of channel geometry and shear stress, in which the computational and experimental design complement one another much more convincingly. For a journal like Micromachines one would expect more focus on the development of the bioprinting technology, hydrogel/material device functionality, or an application involving the cells, not a paper primarily focused on CFD.

3.      The Authors propose modelling the channel with no-slip conditions as a means of simulating the serpentine channel wall elasticity- this argument is unclear and the authors should elaborate on that modelling choice, and ideally validate that hypothesis with experimental data, such as in the form of tracer particles.   

4.      Missing critical materials and methodology. E.g. Details of the meshing are missing from the COMSOL modelling section, along with all details of the hydrogel composition, material mechanical testing, fluorescence microscopy, and cell viability testing. No characterization of the fabricated channels, for instance channel width uniformity would be vital to understanding whether the computational and experimental models are consistent with one another.

5.      Assessment of cell viability. No details of the assay used, or method are provided in the materials and methods section. In the results section, lines 245-249 the authors state the dye binds to both live and dead cells, so it is not a true live-dead assay. Furthermore, it’s likely that some dead cells would have detached from the channel and been washed away. The cell density in figure 8 is relatively low and without a comparison to day 0 no assessment can be made. It also seems unlikely from the cell density that the cells formed an endothelial monolayer. 

Minor:

1.      Figure 1. Legend should be more descriptive and text bigger. Annotating the trace with 0.3T, 0.5T and T would help make Figures 4-7 more intuitive.

2.      Figures 4-7 should have units on the legend.   

3.      Since the fluid is modelled as Newtonian, shear rate and wall stress are linearly related so fewer plots could provide the same information. 

4.      Sentence structure and grammar make the manuscript difficult for readers.

Reviewer 2 Report

The work by Deshmukh describes an exciting approach to simulating, manufacturing, and testing serpentine-based devices that provide a closer representation of in vivo vascular vessels. However, the following major issues need to be addressed prior to publication:

1. The authors claim that they implemented a simulation in COMSOL but the corresponding description of materials and methods is incomplete. For instance, the authors fail to show the governing equations, mesh type and convergence, boundary conditions, etc. Also, the simulation parameters need to be listed in detail.

2. The experimental details of printing are missing, including the materials' properties, which require a throughout characterization via mechanical, spectroscopic, and thermal instruments.

3. The cell seeding protocols and how the authors assured sterility are missing.

4. The authors indicate infusion of the system, but failed to indicate the corresponding flow rates and the infusion protocol.

5. Besides the uncertainty analysis, the authors need to conduct a sensitivity analysis.

6. The results failed to include any of the results found for the cells.

7. An image of the cells is shown in the results but the scale bar is missing and as discussed above none of the cell protocols were shown. A control is missing where cells are grown on a more rigid substrate so that the advantages of the new system are clearly shown to the readers.

8. Overall, the manuscript has potential but it's still in the early stages.